# Analytical Method for Assessing Stability of a Counterbalanced Forklift Truck Assembled with Interchangeable Equipment

**Leonardo Vita ***  and **Davide Gattamelata**

Department of Innovation Technologies, National Institute for Insurance against Accidents at Work (INAIL), Monte Porzio Catone, 00078 Rome, Italy

\* Correspondence: l.vita@inail.it; Tel.: +39-0694181566

**Featured Application: The proposed analytical method is suitable to check the stability (lateral and longitudinal) of the combination of a counterbalanced forklift truck with an interchangeable equipment and to assess if the interchangeable equipment could be safely assembled to the forklift truck.**

**Abstract:** Counterbalanced forklift trucks (FLT) are frequently used in combination with interchangeable equipment in order to handle loads in different manners. The main risks which may arise after assembling interchangeable equipment to a FLT are related to the loss of stability of the assembly. Actually, the presence of interchangeable equipment and the associated payload may change in a significant way the overall centre of gravity with respect to the FLT in its basic configuration with forks. Therefore, the stability limits of the assembly, based on the same footprints on the ground of the FLT alone, are affected by the position of the overall centre of gravity. Thus, the presence of interchangeable equipment could reduce the functionality (e.g., lifting capability, lifting height, etc.) of the FLT in order to continue its stability during use. Often, interchangeable equipment is placed on the market by manufacturers other than the FLT manufacturer. In these cases, the correct and safe coupling of the interchangeable equipment with the FLT is the responsibility of the manufacturers of interchangeable equipment, including the stability risk assessment. Thus, the interchangeable equipment manufacturer should have access to the relevant information of the FLT concerning operative and structural features and its configuration as a procedure for assessing the correct and safe coupling. Otherwise, he should perform experimental stability tests for each model of FLT so that its interchangeable equipment can be fitted. Specific research activity is developed in order to define an analytical procedure to assess the stability of FLT when assembled with interchangeable equipment. Specific typologies of FLTs and interchangeable equipment have been selected in order to better characterise the case study. The analytical equations mimic the static stability tests. The results achieved have been compared to experimental data in order to optimise the procedure. The results attained by the application of the analytical procedure to all the combinations of main typologies of FLTs and the interchangeable equipment selected showed good agreement with experimental tests.

**Keywords:** analytical procedure; forklift truck; interchangeable equipment; static stability assessment

## 1. Introduction

The forklift truck (FLT) is one of the most popular in-plant transport machinery. It also represents a substantial part of the severe occupational injury problem in the manufacturing, agricultural, and transport industries. Many studies focus their attention on FLT in order to identify the main safety issues related to its use. In particular, a study conducted in the US in 1998 reported that 95,000 workers were injured, and 100 were killed each year as a result of FLT incidents [1]. The primary cause of fatalities was tip-over. At the beginning of 2000 in Sweden [2] and in Australia [3], similar investigations revealed that the tip-over of FLTs still remains one of the most important causes of accidents among workers. The causes for

a loss of stability in the FLT could be attributed to travel speed and steering angle [4], with respect to lateral stability, overloading [5,6], and human error [7] for longitudinal stability. With respect to this, the presence of the interchangeable equipment and the associated load could change in a significant way the overall centre of gravity position. Therefore, the stability limits of the assembly, based on the same footprints on the ground of the FLT alone, are affected by the presence of the interchangeable equipment and its load. This may lead to the necessity of reducing functionality (e.g., lifting capability, lifting height, etc.) of the FLT in order to continue to grant its stability during use. According to this, it is important to investigate the stability behavior of FLT. It is possible to use analytical [8] or virtual prototyping [4,9–13] approaches to predict the loss of stability of the FLT with additional experimental validation. Moreover, the last decade has registered an increment in the number of research activities and patents on active systems for increasing the stability of the FLT [14]. However, these systems are not yet in production, and the mentioned approaches are not used to determine the effect of interchangeable equipment and its eventual load on the stability of the FLT. In addition, an analytical approach applied to the FLT's stability is not able to take into account the deflection of the FLT's tires, frame, mast, carriages, and forks. For this reason, the stability of FLT is generally calculated by means of experimental tests. The main target of the proposed methodology is to define an analytical method to predict the stability of the FLT in combination with interchangeable equipment and its intended load. In particular, starting from the technical data of FLT, which is easily accessible from its instruction for use, and taking into account the results of the stability tests developed for FLT with a fork by its manufacturer, it is possible to check if the interchangeable equipment may lead to instability during its use while assembled with FLT. In this way, the effects of the aforementioned deflections of the structural elements of FLT and tires are already included in the results of stability tests, and it is not necessary to mimic them. This simplified approach may be used by the manufacturer of the interchangeable equipment as an alternative to experimental tests for its risk assessment with respect to loss of stability in the complex. The results achieved seem to be in good agreement with the experimental tests, even if, in some cases, the analytical procedure appears more restrictive than the experimental one. This encourages researchers to continue this research activity in order to optimise the procedure and make it much more reliable.

## 2. Materials and Methods

In order to assess the stability of FLT, the standard ISO 22915-2 [15] can be used. The standard specifies the tests for verifying the stability of counterbalanced trucks with masts when equipped with fork arms or with load handling attachments. The standard requires that the FLT is positioned on a tilt table in specific configurations (e.g., orientation with respect to the tilt axis, the direction of the test, with or without a load at the load centre of gravity, lift height, etc.). Then, the minimum value of stability to be achieved during the test with the load at the load centre of gravity is specified as follows:

- 4% longitudinal direction during stacking/retrieving with load at maximum lift height and actual capacity less than 5000 kg;
- 3.5% longitudinal direction during stacking/retrieving with load at maximum lift height and actual capacity, not less than 5000 kg;
- 18% longitudinal direction during travelling with load at lift height for travelling (maximum 500 mm);
- 6% lateral direction during stacking/retrieving with load at maximum lift height.

The test load should have a mass that is equivalent to the actual capacity so that the truck can elevate to the corresponding height when acting through the centre of gravity and nominally positioned at the standard load centre distance (see Figure 1) both horizontally from the front face of the fork arm shank, and vertically from the upper face of the fork arm blade.

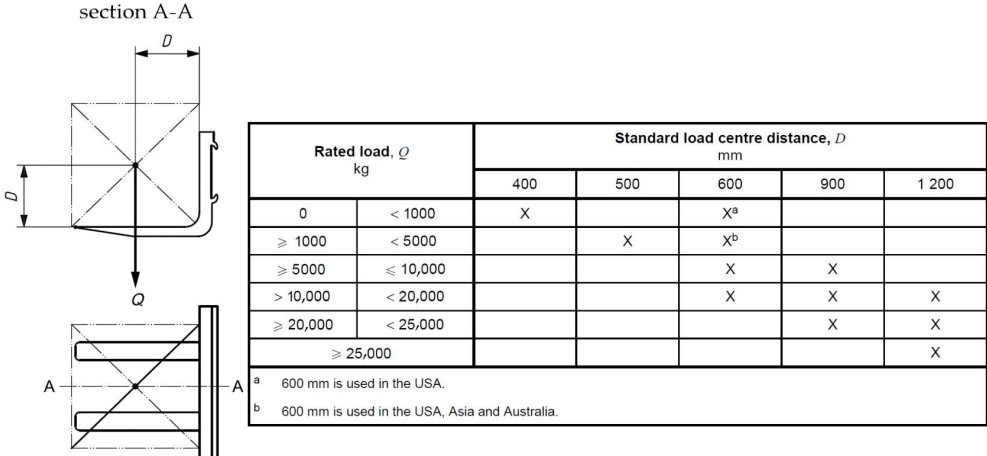

**Figure 1.** Standard load centre distance.

Considering that the additional weight due to the interchangeable equipment itself and the different position of the load centre of gravity might affect the original stability of the truck, the proposed analytical procedure for assessing the stability of FLT in combination with interchangeable equipment is intended to replicate the minimum stability conditions of the FLT as per stability tests in ISO 22915-2. In particular, it is necessary that the longitudinal/transversal moment introduced by the interchangeable equipment does not exceed the longitudinal/transversal moment due to the test load on the fork of the FLT.

### 2.1. Longitudinal Stability

With reference to the longitudinal stability of the FLT in combination with interchangeable equipment, it is necessary to acquire the following data from the instruction handbook and/or the marking plate on the FLT:

- The rated capacity ($Q_{ij}$) with forks according to EN ISO 3691-1 [16]. It can also be defined by means of a load chart depending on the specified height at a specified load centre distance;
- The type of tyres the load chart refers to (if any);
- The fork arm blade thickness(s).
- Similarly, the following data concerning the interchangeable equipment should be acquired in the instruction handbook:
- The total mass of the interchangeable equipment (PCA);
- The distance (L) measured horizontally from the fork carrier to the centre of gravity for the unladen interchangeable equipment;
- The distance ($D_{ai}$) measured horizontally from the fork carrier to the centre of gravity of the load;
- The maximum load permitted by the manufacturer ($Q_i$).

Once known, the aforementioned parameters are necessary to calculate the maximum operating longitudinal moment (OLM) according to each value of the actual capacity ($Q_{ij}$) of the FLT and with respect to the lift height ($H_j$), as follows (see Figure 2):

$$OLM = Q_{ij} \times D_i, \tag{1}$$

where $D_i$ is the sum of the fork arm blade thickness(s) and the specified load centre distance.

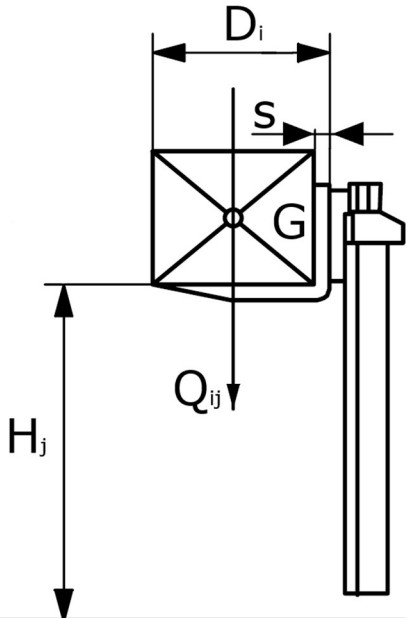

**Figure 2.** Evaluation of OLM: nomenclature. Key: $D_i$—load centre distance from the fork carrier; $H_j$—lift height; $Q_{ij}$—rated capacity according to $D_i$ and $H_j$; s—fork arm blade thickness.

On the other hand, for interchangeable equipment, it is necessary to calculate:

a.    The maximum load at the fork carrier (CI) as the sum of the maximum load permitted by the manufacturer ($Q_i$) and the total mass of the interchangeable equipment (PCA);

b.    The overall longitudinal moment at the fork carrier (MLA) according to Equation (2) and nomenclature in Figure 3.

$$MLA = PCA \times L + Q_i \times D_{ai}. \tag{2}$$

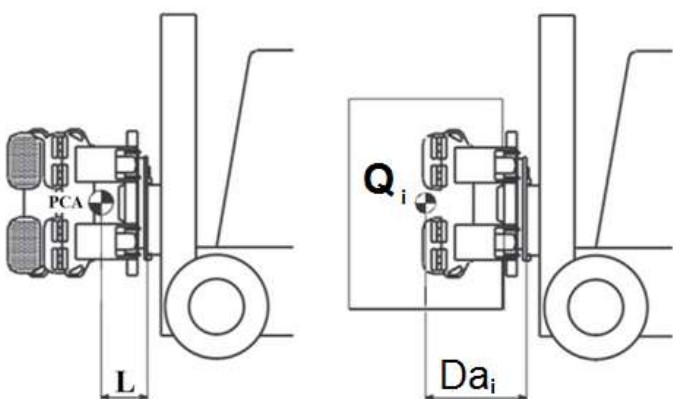

**Figure 3.** Evaluation of MLA: nomenclature. Key: PCA—total mass of the interchangeable equipment; L—distance measured horizontally from the fork carrier to the centre of gravity of the unladen interchangeable equipment; $Da_i$—distance measured horizontally from the fork carrier to the centre of gravity of the load; $Q_i$—maximum load permitted by the manufacturer at $Da_i$.

The last step is to verify for each condition the lift height and load distance which is foreseen in the FLT load chart as:

$$CI \leq Q_{ij}, \tag{3}$$

and

$$MLA \leq OLM. \tag{4}$$

Thus, the actual capacity of the FLT, when combined with interchangeable equipment, is the maximum load that simultaneously satisfies Equations (3) and (4). For this purpose, it could be necessary to limit the value of the maximum load which can be handled with interchangeable equipment. If there are no values of the loads which simultaneously satisfy Equations (3) and (4) for any combination of the lift height and load distance, then the interchangeable equipment is not compatible with the FLT.

### 2.2. Lateral Stability

With reference to the lateral stability of the FLT, and in combination with interchangeable equipment, it is necessary to acquire the following additional data from the instruction handbook and/or the marking plate on the interchangeable equipment:

- The maximum lateral displacement S [mm];
- The total mass (PCAS) of the elements FOR the interchangeable equipment which takes part in the lateral displacement [kg].

On the basis of the acquired data, it is possible to:

- Evaluate the maximum operating transversal moment (OTM) of the truck as per Equation (5) according to the lift height ($H_j$ in Figure 2). With reference to Figure 4, the load is supposed to be on the fork arm of the FLT and is positioned according to the test condition for lateral stability (Test 3) required by ISO 22915-2 at a slope of 6% ($\alpha = 3.4°$).

$$OTM = Q_{ij} \times (S + H_j \sin\alpha). \tag{5}$$

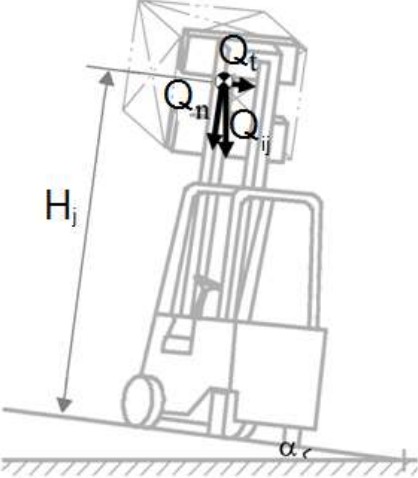

**Figure 4.** Evaluation of OTM: nomenclature. Key: $H_j$—lift height; $Q_{ij}$—rated capacity according to $D_i$ and $H_j$; $Q_n$—rated capacity component normal to tilting platform; $Q_t$—rated capacity component tangent to tilting platform; $\alpha$—slope.

- It is important to evaluate the total transversal moment of the interchangeable equipment (MTA) according to the lift height ($H_j$) as per Equation (6). The FLT is supposed to be positioned according to the test condition for lateral stability required by ISO 22915-2 (Test 3) at a slope of 6% ($\alpha = 3.4°$). The value of lateral displacement should be the maximum allowed by the interchangeable equipment by the nature of the load being handled or by the load handle device employed. The total load at the interface ($CI_L$) should be evaluated as the sum of the maximum load permitted by the manufacturer ($Q_i$ in Figure 3) and the total mass of the elements of the interchangeable equipment which take part in the lateral displacement ($PCA_S$).

$$MTA = CI_L \times S + CI_L \sin\alpha \times H_j. \tag{6}$$

The last step is to verify for each condition the lift height and load distance as foreseen in the FLT load chart:

$$MTA \leq OTM. \tag{7}$$

Thus, the actual capacity of the truck coupled with the interchangeable equipment should be evaluated by considering the loads which allow Equation (7) to be satisfied. For this purpose, it could be necessary to reduce the value of the maximum load permitted for the interchangeable equipment ($Q_i$).

Interchangeable equipment which does not satisfy Equation (7) for any combination of loads is not compatible with FLT.

### 2.3. Lateral Displacement

The interchangeable equipment could be provided with a powered load-handling device, such as a side-shift, which can displace the centre of gravity to a substantial, predetermined extent from the longitudinal centre plane of the FLT. It is also possible that the centre of gravity of the load could be substantially offset from the longitudinal centre plane of the FLT because of the nature of the load being handled or by the load handle device employed. In both cases, a displacement is considered to be substantial when it is more than (Figure 5):

- At 100 mm for FLT with a rated capacity of <5000 kg;
- At 150 mm for FLT with a rated capacity of ≥5000 kg and ≤10,000 kg;
- At 250 mm for FLT with a rated capacity of >10,000 kg and <20,000 kg;
- At 350 mm for FLT with a rated capacity of ≥20,000 kg.

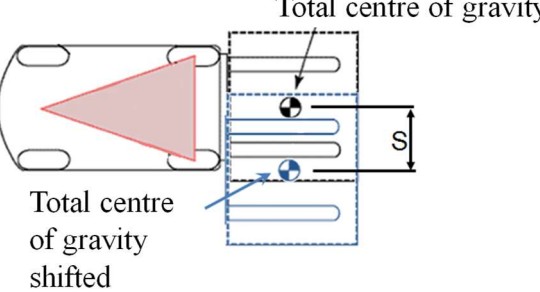

**Figure 5.** Lateral displacement.

Thus, for Equation (5), the value of lateral displacement S to be used should be equal to the value in the above list according to the FLT-rated capacity. For Equation (6), the value of lateral displacement S to be used should be equal to the value in the above list according to the FLT-rated capacity or greater in the event that the interchangeable equipment undergoes substantial lateral displacement. In this case, the value has to be the one defined by the manufacturer of interchangeable equipment or the maximum lateral displacement achieved by the interchangeable equipment. In any case, it should be more than the value listed above according to the FLT-rated capacity.

### 2.4. Rated Capacity of the Assembly

The analytical procedure for evaluating the rated capacity of the assembly (i.e., FLT + interchangeable equipment) is based on Equations (3), (4), and (7). With reference to the flowchart in Figure 6, as a first attempt, the maximum load permitted for the interchangeable equipment ($Q_i$) should be used to check if the longitudinal stability is satisfied according to Equations (3) and (4). If not, the value of the maximum load permitted for the interchangeable equipment should be reduced ($Q_i^I$) until the aforementioned equations are both fulfilled. Then, it is possible to check the lateral stability by means of Equation (7). If the updated maximum load ($Q_i^I$) also fulfills this last equation, it represents the rated capacity of the assembly. Otherwise, it should be reduced until Equation (7) is also fulfilled.

The final value ($Q_i^{II}$) is the rated capacity of the FLT combined with the interchangeable equipment.

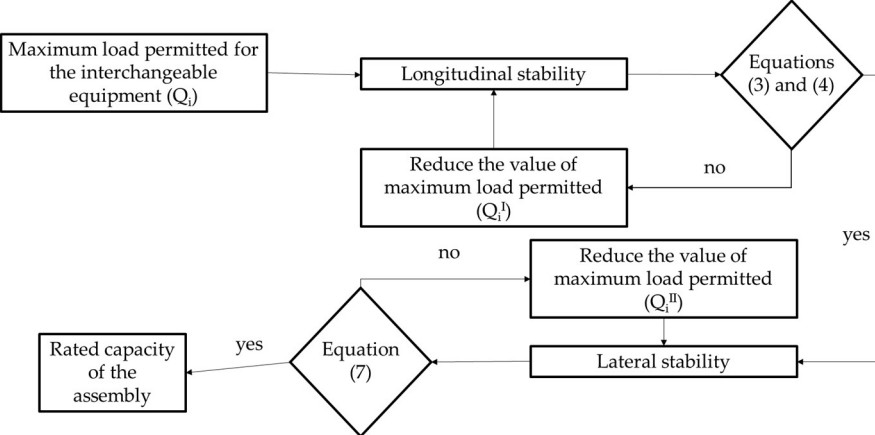

**Figure 6.** Analytical assessment of assembly-rated capacity.

## 3. Results

In order to validate the proposed analytical method, experimental tests were conducted. In particular, a total of six models of FLT and three interchangeable pieces of equipment were selected. The FLTs had a rated capacity of 2 ÷ 2.5 tons, a maximum lift height of 3 ÷ 5 m, and a load centre of gravity at a horizontal distance of 500 mm from the front face of the fork arm shank. With reference to Figure 7, the selected interchangeable equipment included:

- The lateral displacement—is used for the lateral displacement of the fork arms of a predetermined length. It has a rated capacity of 2500 kg, a horizontal distance of the load centre of gravity from the fork carrier of 600 mm, a weight of 55 kg, a horizontal distance of the interchangeable equipment centre of gravity from the fork carrier of 37 mm, and a lateral displacement of 100 mm.
- The clamp with parallel arms—it is used for handling cardboard bales, wool, garbage, reels of cellulose, foam blocks, concrete blocks, kegs, reels, etc. It allows the clamp to load by means of the parallel motion of its forks/arms. It has a rated capacity of 1900 kg, a horizontal distance of the load centre of gravity from the fork carrier of 637 mm, a weight of 448 kg, a horizontal distance of the interchangeable equipment centre of gravity from the fork carrier of 249 mm, and a lateral displacement of 313 mm.
- The clamp for reels—it is used for handling paper reels of a different nature. This attachment can also continuously rotate the load. It has a rated capacity of 2000 kg, a horizontal distance of the load centre of gravity from the fork carrier of 852 mm, a weight of 620 kg, a horizontal distance of the interchangeable equipment centre of gravity from the fork carrier of 235 mm, and a lateral displacement of 425 mm.

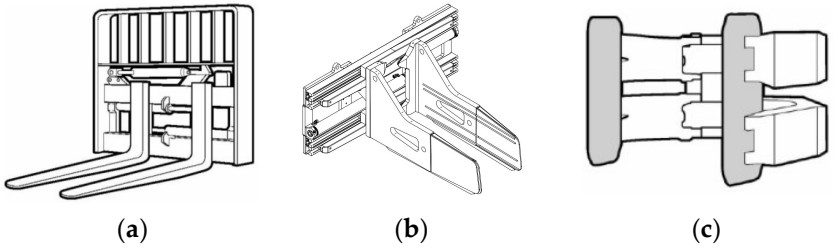

**Figure 7.** Interchangeable equipment: (**a**) Lateral displacement, (**b**) Clamp with parallel arms, and (**c**) Clamp for reels.

For each combination of the aforementioned FLTs and interchangeable equipment, the rated capacity of the assembly (FLT + interchangeable equipment) was evaluated by means

of the analytical procedure that has been compared to the one obtained by the manufacturer of the FLT by means of experimental stability tests. The results are summarized in Table 1 for lateral displacement, in Table 2 for the clamp with parallel arms, and in Table 3 for the clamp for reels.

**Table 1.** Comparison between experimental tests and analytical procedure for lateral displacement.

| FLT | Experimental Test [kg] | Analytical Method [kg] | Delta [%] |
|-----|-----|-----|-----|
| 1 | 2300 | 2195 | −4.5 |
| 2 | 2400 | 2195 | −8.5 |
| 3 | 1740 | 1785 | +2.5 |
| 4 | 1790 | 1785 | −0.3 |
| 5 | 1840 | 1785 | −3.0 |
| 6 | 2310 | 2235 | −3.2 |

**Table 2.** Comparison between experimental tests and analytical procedure for clamp with parallel arms.

| FLT | Experimental Test [kg] | Analytical Method [kg] | Delta [%] |
|-----|-----|-----|-----|
| 1 | 1840 | 1802 | −2.1 |
| 2 | 1900 | 1802 | −5.4 |
| 3 | 1510 | 1502 | −0.5 |
| 4 | 1580 | 1502 | −4.9 |
| 5 | 1610 | 1517 | −5.8 |
| 6 | 1900 | 1942 | +2.2 |

**Table 3.** Comparison between experimental tests and analytical procedure for clamp for reels.

| FLT | Experimental Test [kg] | Analytical Method [kg] | Delta [%] |
|-----|-----|-----|-----|
| 1 | 1440 | 1515 | +5.2 |
| 2 | 1630 | 1515 | −7.0 |
| 3 | 1170 | 1080 | −7.7 |
| 4 | 1170 | 1080 | −7.7 |
| 5 | 1240 | 1095 | −11.7 |
| 6 | 1630 | 1410 | −13.5 |

## 4. Discussion

Basically, when an analytical model is developed, it is used to predict the stability factor of a basic machine. Thus, static and, in some cases, dynamic forces acting on the basic machine while facing an overturn are considered [17–20]. For instance, in [21], the zero moment point theory to calculate the maximum lifting capacity of the truck crane when considering the dynamic load is described in order to predict the overturning state of the crane and reduce its risk in operation. When referring to FLT, it has to be considered that the dynamic effects due to the load are minimised. In fact, the load is stabilised (e.g., by means of a pallet or similar devices) on the forks. However, the maximum lift capacity of a FLT is always based on its stability. Thus, knowing the correct location of its centre of gravity is crucial. In [22], a method for determining the center of gravity of a forklift truck is illustrated. A similar approach is described in [23], which points out in the discussions that the deflection of tires, frames, masts, carriages, forks, carriage looseness, and mast looseness have an effect on the stability of a fork truck and are virtually impossible to solve mathematically. Therefore, using a tilting platform for conducting stability tests would be realistic because, in this way, it is possible to simulate the effect of centrifugal forces that take place when maneuvering, and it automatically takes into consideration the

aforementioned factors that affect the stability of a FLT. Standard ISO 22915-2 defines the stability tests that need to be performed. With reference to the aforementioned standard, in [8], the stability equations were derived with detailed derivations for the four kinds of stability tests. The effects of acceleration and centrifugal forces were also additionally included in the equations. However, according to the authors' knowledge, at present, the analytical procedures that are at one's disposal refer to FLTs with forks. No analytical procedures are available to evaluate the stability of FLT when combined with interchangeable equipment. A useful guide for identifying interchangeable equipment can be found in [24], where the European materials handling federation (FEM) proposes a classification for the attachments of the FLT with reference to the applicable legislation and on the basis of the documentation available to date and the technical knowledge of its members (industrial trucks manufacturers and attachment manufacturers). Hence, the analytical method herein proposed is focused on filling this gap. It is mainly based on the static equilibrium (force and moment) of a rigid body. Even if this simplified approach is not able to mimic the flexibility of the FLT structure and the dynamic effects of the lifting operations, it is able to capture the changes introduced by the interchangeable equipment in the rated capacity of the FLT with forks. The proposed analytical approach takes into account both the fact that FLT has to fulfill stability requirements defined in ISO 22915-2 and that the actual capacity at a maximum lift height with the load centre distance should accordingly be provided by the FLT's manufacturer. Thus, the main target is to verify if the centre of gravity of the interchangeable equipment plus the centre of gravity of the maximum intended load remains within the limits of the actual capacity of the FLT. If so, no changes are needed, and the FLT could be used with the interchangeable equipment and its maximum intended load according to the rated capacity of FLT with forks. If not, it is necessary to define which is the maximum load that the FLT may lift with the interchangeable equipment (i.e., the rated capacity of the assembly) that is equivalent to the rated capacity of FLTs with forks. As a consequence, it is not necessary to model the FLT (basic machine) in order to define its static stability limit. In fact, the main assumption is that the FLT is stable, at least at the slope required by the ISO 22915-2 standard, which should be experimentally verified by the FLT manufacturer. Thus, it is necessary to model only the deviation of FLTs to the centre of gravity when introduced by interchangeable equipment and its intended load.

According to the proposed analytical procedure, the rated capacity of the assembly is determined as the maximum value, which allows for the satisfaction of the contemporary Equations (3), (4), and (7). The analytical results summarised in Tables 1–3 are compared to the experimental tests developed by the manufacturer of the FLT assembled with the same interchangeable equipment. With the exception of FLT numbers five and six combined with the clamp for reels, where the most severe condition is the lateral stability, for all the other combinations, the longitudinal stability represents the worst-case condition. The rated capacity, evaluated by means of the analytical method, is, in most of the test cases, lower than the one obtained by the means of experimental tests. This is mainly due to the assumption of rigid bodies in the analytical model. However, a more restrictive value in terms of the rated capacity is in favor of safety. The average percentage of rated capacity reduction is 5.7, which is not so relevant in terms of FLT functionality. However, it is necessary to observe that three analytical results on eighteen are higher than the experimental ones. Considering that each of them refers to different FLTs in combination with different interchangeable equipment, it is not possible to clearly identify the parameter which may affect the results. In addition, it should be considered that even experimental tests are affected by boundary conditions, which may affect the experimental result. At this stage of the study, it is not possible to replicate them in a simple analytical model. In addition, the maximum delta registered on rated capacity is about +5%, which is in line with the tolerance of the instruments for detecting the static stability angle on the experimental test rig. The comparison with experimental tests encourages us to use this simplified analytical procedure as an alternative to extensive experimental tests on the FLT assembled with interchangeable equipment. In fact, it is able to capture the main changes

in the centre of gravity location, which mainly affect the rated capacity of the assembly. Nevertheless, if there are no values of load able to fulfill Equations (3), (4), and (7), it is necessary to pay particular attention. In this case, the combination of the interchangeable equipment and the FLT may not be advisable due to the potential loss of stability during lifting operations.

## 5. Conclusions

The analytical procedure herein presented represents for manufacturers an alternative method to experimental tests in order to define the rated capacity of a FLT assembled with interchangeable equipment. This procedure is easier to perform, less costly, and less time-consuming with respect to experimental tests. The innovative approach relays on the assumption that the FLT with a fork should be stable at specific angles according to which its rated capacity is tested out. Thus, it is not necessary to model all the assemblies but only the variations introduced by the interchangeable equipment and its intended load. Moreover, this procedure is particularly useful for the stability risk assessment of FLTs and interchangeable equipment that are already in use and that are assembled by operators and not by engineers or trained technicians. The results achieved seem to be in good agreement with the experimental tests, even if, in some cases, the analytical procedure seems to reduce the rated capacity of the assembly to a greater extent than the one evaluated by means of experimental tests. However, it should be noted that this reduced rated capacity is in favor of safety. On the basis of the results of this study, the procedure needs to be refined. Increasing the number of numerical applications will allow for a larger set of data to be generated in order to compare the results with the manufacturer's database of the stability test measurements. This will allow us to optimise the procedure and make it much more reliable without increasing the level of information required. Moreover, once the procedure is refined and optimised, it can also be applied in other sectors, such as earth-moving machinery and agricultural machinery.

**Author Contributions:** Conceptualization, L.V. and D.G.; Data curation, L.V. and D.G.; Formal analysis, L.V. and D.G.; Investigation, L.V. and D.G.; Methodology, L.V. and D.G.; Validation, L.V. and D.G.; Writing—original draft, L.V. and D.G.; Writing—review and editing, L.V. All authors have read and agreed to the published version of the manuscript.

**Funding:** This research received no external funding.

**Institutional Review Board Statement:** Not applicable.

**Informed Consent Statement:** Not applicable.

**Data Availability Statement:** Data concerning analytical results and experimental tests will be be provided upon request to the corresponding author.

**Conflicts of Interest:** The authors declare no conflict of interest.

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
