# Peer review of "Analytical Method for Assessing Stability of a Counterbalanced Forklift Truck Assembled with Interchangeable Equipment"

_applsci, doi:10.3390/app13021206_

Round 1

Reviewer 1 Report

1) The problem statement needs to be made clear in the abstract, just before the authors state their aim of study.

2) The problem statement also needs to be clear in the introduction.

3) There are very few scholarly literatures cited in the introduction text, much less updated ones (in years of 2020-2022). This needs to be fixed. 

4) The data analysis between the experimental and analytical results is too elementary. Is there no way to conduct a proper hypothesis test and include analyses such as t-tests (for normalised data) or Mann-Whitney U Test (for non-normal data)? That would improve the conjecture of the study.

5) The discussion is too brief. There is not enough literature cited in the discussion to compare the findings or outcome of the study. This makes the paper less scientific and more like a technical report.

6) The conclusion needs to be compartmentalised into achievement of objectives, major findings, limitations, and directions for future research. 

7) The English needs extensive editing.

Reviewer 2 Report

Dear Authors

your work is about counterbalanced forklift trucks (FLT) that are frequently used in combination with changeable equipment in order to handle loads in different manners. This is an important matter that can be considered also towards new kinds of transportation means. I suggest, in the future, to go on studying this issue, in order to find new apllications.

Best Regards

Reviewer 3 Report

It might help to include center of gravity in the analysis. with and without the load.

Good luck

Round 2

Reviewer 1 Report

The manuscript has been improved. However, I still do not feel that the gap of the study is properly highlighted within the problem statement. If possible, the discussion should cite more papers to triangulate the findings of the study with other studies. 
